# Effect of Combining Low Temperature Plasma, Negative Pressure Wound Therapy, and Bone Marrow Mesenchymal Stem Cells on an Acute Skin Wound Healing Mouse Model

**DOI:** 10.3390/ijms21103675

**Published:** 2020-05-23

**Authors:** Hui Song Cui, So Young Joo, Yoon Soo Cho, Ji Heon Park, June-Bum Kim, Cheong Hoon Seo

**Affiliations:** 1Burn Institute, Department of Rehabilitation Medicine, Hangang Sacred Heart Hospital, College of Medicine, Hallym University, Seoul 07247, Korea; bioeast@hanmail.net (H.S.C.); jiheonpark84@naver.com (J.H.P.); 2Department of Rehabilitation Medicine, Hangang Sacred Heart Hospital, College of Medicine, Hallym University, Seoul 07247, Korea; anyany98@naver.com (S.Y.J.); hamays@hanmail.net (Y.S.C.); 3Department of Pediatrics, Hangang Sacred Heart Hospital, College of Medicine, Hallym University, Seoul 07247, Korea

**Keywords:** wound healing, low-temperature plasma, negative pressure wound therapy, bone marrow mesenchymal stem cell, re-epithelialization

## Abstract

Low-temperature plasma (LTP; 3 min/day), negative pressure wound therapy (NPWT; 4 h/day), and bone marrow mesenchymal stem cells (MSCs; 1 × 10^6^ cells/day) were used as mono- and combination therapy in an acute excisional skin wound-healing ICR mouse model. These therapies have been beneficial in treating wounds. We investigated the effectiveness of monotherapy with LTP, NPWT, and MSC and combination therapy with LTP + MSC, LTP + NPWT, NPWT + MSC, and LTP + NPWT + MSC on skin wounds in mice for seven consecutive days. Gene expression, protein expression, and epithelial thickness were analyzed using real time polymerase chain reaction (RT-qPCR), western blotting, and hematoxylin and eosin staining (H&E), respectively. Wound closure was also evaluated. Wound closure was significantly accelerated in monotherapy groups, whereas more accelerated in combination therapy groups. Tumor necrosis factor-α (TNF-α) expression was increased in the LTP monotherapy group but decreased in the NPWT, MSC, and combination therapy groups. Expressions of vascular endothelial growth factor (VEGF), α-smooth muscle actin (α-SMA), and type I collagen were increased in the combination therapy groups. Re-epithelialization was also considerably accelerated in combination therapy groups. Our findings suggest that combination therapy with LPT, NPWT, and MSC exert a synergistic effect on wound healing, representing a promising strategy for the treatment of acute wounds.

## 1. Introduction 

Wound healing is an important physiological process that restores skin integrity after injury. Normal wound repair is a well-orchestrated consolidation of complex biological and active molecular events, including cell migration, proliferation, differentiation, collagen deposition, angiogenesis, and remodeling [1]. This orderly process is impaired in many pathological conditions, such as diabetic foot, which results in serious consequences [2,3]. Unfortunately, an effective treatment for optimal wound healing in all populations remains elusive. However, low-temperature plasma (LTP), negative pressure wound therapy (NPWT), and bone marrow mesenchymal stem cells (MSCs) are effective and well-studied treatments [4,5,6,7,8,9].

LTP devices generate an ionized gas or plasma composed of hydroxyl radicals and reactive oxygen/nitrogen, under atmospheric pressure and near room temperature, which exerts multiple biological effects, including anticancer, antibacterial, and anticoagulation effects [10,11,12], and, in particular, wound healing. Recent studies have revealed the beneficial effects of LTP via modulating wound-healing processes, such as increasing dermal fibroblast proliferation and migration, and inducing cytokines and growth factors relevant to wound healing in vitro [4,13]. These effects, along with a decreased wound size accompanied by increased α-SMA expression and extracellular matrix deposition, have been further verified using a murine model of dermal full-thickness ear wounds [4,5].

NPWT is an adjuvant therapy using a vacuum dressing system that involves the controlled application of negative pressure to the wound environment through a sealed dressing with a tube connected to a vacuum pump to remove fluid from the wound bed; the microvascular suction force induces tissue perfusion via increased blood flow [7]. NPWT has been well-established in ameliorating wound healing in diabetic foot ulcers, pressure ulcers, or chronic wounds [7,14]. Early clinical evidence suggested that NPWT decreased the levels of tumor necrosis factor-alpha (TNF-α), a proinflammatory cytokine in wound fluid, and induced angiogenesis via increasing angiogenesis-associated growth factors in wound tissues from debilitated patients [6,15]. Moreover, in a mouse model, NPWT promoted tissue granulation and increased cell proliferation and collagen production through the activation of mast cells [16].

MSCs are the main stem cells used for cell therapy, characterized by their self-renewal ability and capacity to differentiate into multiple tissue-forming cell lineages [17]. MSCs can differentiate into multiple skin cell types, including keratinocytes, endothelial cells, and pericytes, which contribute to wound healing [18]. Likewise, another characteristic of MSCs is their strong paracrine property, releasing various cytokines, growth factors, and chemokines to modulate all phases of the wound-healing process, including accelerated wound closure, increased angiogenesis, decreased wound inflammation, and regulation of extracellular matrix remodeling [8,9,19].

Using an acute excisional wound-healing ICR mouse model, we demonstrated combination therapy with LTP, NPWT, and MSCs, which represent a promising strategy in modulating inflammation angiogenesis and granulation tissue formation in order to accelerate wound healing.

## 2. Results

### 2.1. Characterization of MSCs

Bone marrow MSC purity was determined via fluorescence-activated cell sorting (FACS) using cell-surface antigens. FACS showed that the cells strongly expressed positive MSC markers: 97.9% expressed CD44 (receptor for hyaluronate and osteopontin), and 99.2% expressed CD29 (β1 integrin) [20]. FACS showed that the cells weakly expressed negative MSC markers: 0.4% expressed CD11b (macrophage and monocyte marker), 0.5% expressed CD34 (primitive hematopoietic progenitor and endothelial cell marker), and 0.2% expressed CD45 (pan-leukocyte marker) [20] (Figure 1).

### 2.2. Effect of Mono- and Combination Therapy on Wound Closure in ICR Mouse Full-Thickness Excisional Wound Model

We evaluated wound closure as the percentage of wound surface area (WSA) remaining after seven consecutive days of treatment, where day zero indicated 100% WSA. Percentage of surface wound area remaining was significantly (*p* < 0.05) lower for monotherapy: LTP (3 min/day), 26.8%; MSC (1 × 10^6^ cells/day), 24.5%; and NPWT (4 h/day), 24.8% than the untreated control, 53.4% (Figure 2). Combination therapy with LTP + MSC, LTP + NPWT, MSC + NPWT, and LTP + MSC + NPWT improved wound closure by reducing the WSA significantly (*p* < 0.05) to 16.9%, 15.5%, 15.7%, and 9.9%, compared with monotherapy, respectively (Figure 2). 

### 2.3. Effect of Mono- and Combination Therapy on TNF-α and VEGF Expression in Wound Tissue

To determine whether the mono- or combination therapy modulated inflammation, we measured the expression of TNF-α, a proinflammatory cytokine, using RT-qPCR and Western blotting in wound tissues (Figure 3A,B). Monotherapy with LTP significantly (*p* < 0.05) increased the mRNA and protein expression of TNF-α at day seven compared with the control (Figure 3A,B). Monotherapy with NPWT and MSC, and combination therapy with LTP + MSC, LTP + NPWT, MSC + NPWT, and LTP + MSC + NPWT, significantly (*p* < 0.05) decreased the mRNA and protein expression of TNF-α compared to the control (Figure 3A,B). There were no statistically significant differences between monotherapy and combination therapy groups. The expression of VEGF, an angiogenesis marker, in wound tissue was investigated at day seven. Monotherapy with LTP, MSC, and NPWT significantly (*p* < 0.05) increased both the mRNA and protein expression of VEGF compared with the control (Figure 3C,D). Compared to monotherapy, combination therapy with LTP + MSC, LTP + NPWT, MSC + NPWT, and LTP + MSC + NPWT significantly (*p* < 0.05) increased both the mRNA and protein expression of VEGF (Figure 3C,D).

### 2.4. Effect of Mono- and Combination Therapy on α-SMA Expression and Collagen Deposition in Wound Tissues 

α-SMA expression and collagen deposition may predict wound-healing progress. Monotherapy with LTP, MSC, and NPWT significantly (*p* < 0.05) increased both the mRNA and protein expression of α-SMA and type I collagen in wound tissue at day seven compared with the control (Figure 4A,B). Compared with monotherapy, combination therapy with LTP + MSC, LTP + NPWT, MSC + NPWT, and LTP + MSC + NPWT significantly (*p* < 0.05) increased both the mRNA and protein expression of α-SMA and type I collagen (Figure 4A,B).

### 2.5. Effect of Mono- and Combination Therapy on Re-Epithelialization in Wound Tissue

Complete re-epithelialization was observed in all groups on day 21 after wounding. Epidermal thickness evaluated after H&E staining and quantified. Monotherapy with LTP, MSC, and NPWT significantly (*p* < 0.05) accelerated re-epithelialization compared with the control (Figure 5A,B). Furthermore, compared with monotherapy, combination therapy with LTP + MSC, LTP + NPWT, MSC + NPWT, and LTP + MSC + NPWT significantly (*p* < 0.05) accelerated re-epithelialization (Figure 5A,B).

## 3. Discussion

We evaluated the therapeutic effects of combination therapy with LTP, MSCs, and NPWT on wound healing using a full-thickness excisional wound model in ICR mice. Mono- and combination therapy with LTP, MSCs, and NPWT modulated inflammation, enhanced angiogenesis, promoted granulation tissue formation, and accelerated re-epithelialization.

Currently, there is no agreement on the LTP techniques producing optimal therapeutic effects, particularly in relation to wound healing. The duration of treatment sessions using LTP is highly variable among the relevant literature. We found that LTP therapy for three min did not affect fibroblast viability; however, treatment times greater than five min impacted cell survival [13]. Our results suggest three min would be a safe duration in clinical applications. High stem cell purity is necessary, as contamination of the MSC preparation can alter experimental outcomes. The purity of stem cells used in our research was greater than 95% (Figure 1). Standard clinical practice for the application of NPWT to improve wound healing is 2–4 h at 125 mm Hg daily for a period of seven days to 16 months, depending on the location and extent of the injury [21,22,23]. We followed these recommendations and applied NPWT for four h at 125 mm Hg for seven days. 

The beneficial effects of LTP on acute wound healing are mainly attributed to the induction of wound healing-related chemokines, cytokines, and growth factors, including macrophages (MCP-1, IL-6, and TGF-β1); fibroblasts; and keratinocytes [4,5,13]. Accumulation of macrophages and induction of TNF-α expression were observed during the inflammatory phase of wound healing after LTP therapy [4,24]. In accordance with the relevant literature, we found that TNF-α expression peaked on the fifth day after wounding (Appendix A), suggesting that the acute inflammatory phase is promoted by LTP therapy. TNF-α exerts an angiogenic effect during wound healing through the induction of hypoxia-inducible factor (HIF)-1α, an upstream regulator of angiogenesis [25]. Induction of the TNF-α expression induced the TNF-stimulated gene 6 (TGS-6) antioxidative defense system and increased the production of TSG-6 (Appendix A), an anti-inflammatory protein, which inhibits macrophage activation and inflammation and accelerates wound healing [25]. This was in contrast to therapy with NPWT, which decreased inflammation by reducing the TNF-α, IL-6, and iNOS levels [15]. MSCs have an anti-inflammatory effect, mainly attributed to the release of TNF-α inhibiting TSG-6 [26]. Collectively, this supports our results, indicating combination therapy with LTP + NPWT or LTP + MSC reduced TNF-α expression in wound tissue (Figure 3A,B).

We observed increased VEGF expression after mono- and combination therapy with LTP, NPWT, and MSCs (Figure 3C,D). Previously, we reported that LTP treatment induced the secretion of multiple angiogenic growth factors, including VEGF, platelet-derived growth factor (PDGF), and angiopoietin, through the upregulation of HIF-1α in dermal fibroblasts and keratinocytes [13,27]. In rabbits, NPWT induced HIF-1α and VEGF expression in wound tissues [28]. BM-MSC treatment significantly increased VEGF expression in wounds via the paracrine effect, and BM-MSCs were detected near, but not in, vascular structures [29].

During the wound-healing process, the fibroblasts from the wound edge migrated to the wound bed, where they proliferated and differentiated into α-SMA-expressing myofibroblasts. The myofibroblasts then secreted and deposited extracellular matrix (ECM) components such as collagen and initiated wound contraction [30]. Therefore, α-SMA is a marker of mature myofibroblasts and indicates the phase of healing. LTP and NPWT therapy induced the production of TGF-β1 and increased the expression of α-SMA and type I collagen in the fibroblasts of wounded tissues from animals [4,31,32]. TGF-β1 is involved in several wound repair processes, including modulating inflammation, stimulating angiogenesis, re-epithelization, fibroblast proliferation, collagen synthesis, and ECM deposition and remodeling [33]. Interestingly, it reduced TGF-β1 levels in the dermal component, while high levels of TGF-β1 are detected in the hyperkeratotic epidermis surrounding the chronic ulcer [34]. MSCs can increase the expression of α-SMA and collagen in fibroblasts and wound tissues through paracrine mechanisms, which involve the release of numerous soluble growth factors related to wound healing [35,36]. α-SMA and type I collagen contribute to wound contraction, which is caused by myofibroblast migration in granulation tissue, pulling the wound edges toward the center of the wound [30]. Additionally, the compaction of collagen is a major force in wound contraction [30]. Therefore, increased expression of α-SMA, a myofibroblast marker, and type I collagen are indicative of wound contraction (Figure 2 and Figure 4).

Re-epithelialization is a decisive parameter when evaluating successful wound closure. We demonstrated that LTP, NPWT, and MSC monotherapy promoted the epithelialization of wound tissues compared to an untreated control. Moreover, combination therapy produced greater epithelialization than monotherapy, which indicated that re-epithelialization contributed to faster wound healing. The migration of keratinocytes is critical to wound re-epithelialization, and in the absence of this process, wounds will not heal [37]. In vitro LTP therapy enhanced the migration of keratinocytes in humans and NPWT-accelerated monolayer keratinocyte migration via the activation of cdc42-dependent signaling pathways [27,38]. Moreover, gene analysis in human biopsies showed that NPWT highly induced changes in the expression of several genes, including IL-8; CXCL5; and MMP-1, 3, and 10, which are associated with keratinocyte migration [39]. In vitro scratch assays showed a substantially increased migration of keratinocytes in an MSC-conditioned medium through interactions with trophic factors, including TGF-β1, secreted from MSCs [40]. 

LTP is mainly focused on dermatological applications aiming to enhance tissue regeneration for rapid and optimal wound healing or treat skin diseases [41]. Studies exploring cell-based wound repair have mostly focused on the MSC transplantation via the direct injection of cells into the edges of the wound. This treatment has been demonstrated effective and safe for acute wound healing in clinical trials [42]. Twenty years of research, including data from large mechanistic clinical trials, have demonstrated NPWT’s ability to treat complex wounds. Combination therapy showed significant therapeutic benefits in a full-thickness skin wound model using ICR mice; however, owing to significant differences in the anatomy and physiology of human skin, limitations apply for translating our results on wound healing in humans. Wound healing in mice is mediated by the panniculus carnosus muscle, which, importantly, is structurally and functionally different in lower mammals than in humans [43]. Nevertheless, mouse models can help us to understand the diseases of the skin and predict treatment efficacy. Hence, for combination therapy with LTP, NPWT, and MSCs for complex wound treatment as a simple, feasible, time-saving method, its outcome in humans is still a potential prospect.

## 4. Materials and Methods

### 4.1. Isolation and Purification of Bone Marrow Mesenchymal Stem Cells (MSCs)

Female ICR mice (6–8 weeks of age) were purchased from Korean Animal Technology (Koatech, Pyeongtaek, Korea). Mice were sacrificed by cervical dislocation, and bone marrow was harvested by flushing the bone marrow from the femur and tibia with MSC culture medium, minimum essential medium eagle, alpha modification (α-MEM; Welgene, Daegu, Korea), supplemented with 20% fetal bovine serum (FBS; Biowest, Riverside, MO, USA) and 1% antibiotics—antimycotics (Gibco, Life Technologies, Carlsbad, CA, USA). After centrifugation at 1500 rpm for 5 min, the pellet was suspended in Dulbecco’s phosphate-buffered saline (DPBS), which was then mixed with lysing buffer (BD Biosciences, San Diego, CA, USA) at a 1:3 ratio to lyse the red blood cells, followed by gentle vortexing at room temperature (RT) protected from light for 15 min. Centrifugation was repeated at 1500 rpm for 5 min, and the pellet was resuspended in MSC culture medium cultured at 37 °C and 5% CO_2_ atmosphere. Nonadherent cells were carefully removed by DPBS washing, and the culture medium was changed every 2 to 3 days. When the cells reached 90% confluence at passage 1, they were harvested by the treatment cell detachment solution Accutase^®^ (Thermo Fisher Scientific, Waltham, MA, USA). MSCs were purified by negative selection using a mouse hematopoietic progenitor cell isolation kit (Stem Cell Technologies, Vancouver, Canada) against CD3, CD11b, CD19, CD45R/B220, Ly6G/C (Gr-1), and TER119 and a mouse mesenchymal progenitor enrichment kit (Stem Cell Technologies, Vancouver, Canada) against CD45. Cells from passages 2 to 4 were used for animal experiments.

### 4.2. Flow Cytometry

To confirm the purity of the MSCs, flow cytometry was conducted at Yonsei Medical Research Center (Seoul, Korea). MSCs from passage 2 were resuspended in a flow cytometry staining buffer (eBioscience, San Diego, CA, USA) at 1 × 10^7^ cells/mL. Aliquots of 50 μL were incubated in a microtube at 0.5 ug/test concentration with monoclonal antibodies (Table 1). All antibodies were purchased from eBioscience, Inc. Cells were washed twice with a flow cytometry staining buffer and then transferred to a FACS tube (BD Biosciences, San Diego, CA, USA) before analysis on a FACSVerse flow cytometer (BD Biosciences). 

### 4.3. LTP Treatment

LTP was applied as described in our previous study [13]. A sinusoidal voltage of 5.99-kV, frequency of 13.0 kHz, and electric power of 42 W were applied to generate a working gas mixture of air (50 cm^3^/min) and He (5000 cm^3^/min, 99.99% purity). LTP treatment to wounds was applied daily for 3 min at a distance of 3 cm from the wound surface for 7 days.

### 4.4. NPWT Treatment

The NPWT device was the CuraVAC™ Ag (Daewoong Pharm Co, Ltd., Seoul, Korea), which has a foam dressing that contains silver nanoparticles. The following procedure was applied to the mice wounds: First, a sterile foam dressing was placed on the wound. Next, a circular hole was cut in the Tegaderm^TM^ (3M, Saint Paul, MN, USA), and a silicone suction head was placed through the hole. Then, the foam and an additional 2–3 cm of surrounding intact skin were covered. Finally, the suction head tube was connected to the vacuum pump. Negative pressure was applied via the therapy unit, causing the dressing to collapse into the wound. The negative pressure was maintained for 4 h at 125 mm Hg, and NPWT was applied for 7 days. 

### 4.5. MSC Therapy 

MSCs were suspended in DPBS (1 × 10^7^ cells/mL), and 100 μL (1 × 10^6^ cells) was subcutaneously injected into the wound edge daily for 7 days.

### 4.6. Wound Model and Treatment

Mice were anesthetized and maintained using 100% oxygen with 2.5% isoflurane (Hana Pharm, Seoul, Korea) during the procedure. The backs of mice were shaved and sterilized with 70% alcohol, and a full-thickness dorsal wound 12 mm in diameter was created on each mouse using a 12-mm dermal biopsy punch (Acuderm, Fort Lauderdale, FL, USA) on the back of the mice. Subsequently, the mice were randomly classified into eight groups: (1) Wounds allowed to heal naturally without any treatment (control group). (2) Wounds treated with LTP for 3 min for 7 days (LTP group). (3) Wounds treated with NPWT for 4 h at 125 mm Hg per day for 7 days (NPWT group). (4) Wounds subcutaneously injected with 1 × 10^6^ of MSCs for 7 days (MSC group). (5) Wounds treated with LTP followed by NPWT daily for 7 days (LTP + NPWT group). (6) Wounds treated with LTP followed by MSC therapy daily for 7 days (LTP + MSC group). (7) MSC application followed by NPWT therapy for 7 days (NPWT + MSC group). (8) Wounds treated with LTP followed by MSC, and then NPWT therapy for 7 days (LTP + MSC+ NPWT group). The daily wound dressing was performed using Tegaderm^TM^ (3M) during the experimental period. All animal experiments were performed in compliance with the Guide for the Care and Use of Laboratory Animals of the National Institutes of Health. This study was approved by the Animal Research Ethics Board of Hallym University (HMC2017-2-1121-30, 19/12/2017).

### 4.7. Measurement of Wound Closure

To analyze the wound size, images were obtained using a digital camera (Nikon, Tokyo, Japan) on days 0 and 7 after wound formation. Quantification of wound closure was performed using ImageJ software (National Institutes of Health, Bethesda, MD, USA) normalized with the wound size at day 0 as 100% (https://imagej.nih.gov/ij).

### 4.8. Quantitative Real-Time PCR (RT-qPCR)

To isolate total RNA from the wound tissue, samples were homogenized in Trizol (Invitrogen, Carlsbad, CA, USA) using gentleMACS™ Dissociator (Miltenyi Biotect, Bergisch-Gladbach, Germany), and total RNA was extracted using a ReliaPrepTM RNA Miniprep system (Promega, Madison, WI, USA). RNA concentration was measured using a nanodrop spectrophotometer (BioTek Instruments Inc., Winooski, VT, USA), and 1-µg RNA was converted to cDNA using PrimeScript™ RT master mix (Takara Bio Inc., Kusatsu, Japan). RT-qPCR was performed on a Light Cycler 480 system (Roche, Basel, Switzerland) using PCR premix (Takara Bio Inc.), 50-ng cDNA, and 0.5-µM primers (Table 2). The mRNA expression level for each target gene was normalized to the expression of glyceraldehyde-3-phosphate dehydrogenase (GAPDH) by the 2^−ΔΔCt^ method [44]. 

### 4.9. Western Blot

To obtain the total protein content from the wound tissue, samples were homogenized using gentleMACS™ Dissociator (Miltenyi Biotect) in an RIPA buffer (Biosesang, Seongnam, Korea), which was added to the protease (Sigma-Aldrich, Saint Louis, MO, USA) and phosphatase inhibitors (Roche). The samples were agitated for 1 h at 4 °C and then centrifuged for 30 min at 15,000 rpm and 4 °C. The protein concentration of the clear supernatant was measured by a BCA kit (Thermo Fisher Scientific, Carlsbad, CA, USA), and the supernatant was mixed with 2× Laemmli buffer (Thermo Fisher Scientific) before boiling at 95 °C for 5 min. Then, proteins were separated (30-μg protein/well) using 8% SDS-PAGE gel and electrotransferred onto preactivated PVDF membranes (Merck Millipore, Billerica, MA, USA). The membranes were blocked with 5% skim milk for 1 h at RT and then incubated overnight with monoclonal mouse anti-TNFα (1:1000; Thermo Fisher Scientific,), polyclonal mouse anti-αSMA antibody (1:500; Abcam, Cambridge, UK), polyclonal rabbit anti-collagen-I antibody (1:1000; Abcam), polyclonal rabbit anti-VEGF antibody (1:1000; Abcam), and polyclonal rabbit anti β-actin antibody (1:3000; Cell Signaling, Danvers, MA, USA). The membranes were washed 3× (10 min/wash) with Tris-buffered saline and 0.1% Tween 20 (TBST) buffer and then incubated with peroxidase-conjugated anti-mouse IgG antibody or anti-rabbit IgG antibody (1:3000; Merck Millipore) for 2 h at RT. They were then washed 3× (10 min/wash) and developed with an ECL detection kit (Thermo Fisher Scientific). The band images were created using a chemiluminescence imaging system (Atto, Tokyo, Japan), and the band densities were analyzed with CS Analyzer 4 software (Atto). 

### 4.10. Measurement of Epithelial Thickness

Mice were euthanized by inhalation of CO_2_ gas at day 21 after wounding, and tissue samples were excised, including the complete epithelial region around the wound. The tissue samples were then immediately fixed in 4% neutral-buffered formalin overnight at 4 °C and placed in 30% sucrose for 72 h. Subsequently, they were embedded using an O.C.T. compound (Sakura Finetek, Torrance, CA, USA) in cryomolds, placed in liquid nitrogen, and stored in a deep freezer until use. The cryo samples were cut into sections 10 μm thick using a cryostat (Leica Biosystems, Nussloch, Germany). After drying at RT, the sections were fixed in cold methanol, washed for 5 min with PBS, stained for 10 min with Mayer’s hematoxylin (Cancer Diagnostics, Inc., Durham, NC, USA), and then stained for 30 s with eosin solution (Cancer Diagnostics, Inc.) at RT. Finally, they were cover-slipped with an aqueous mounting medium (Dako, Santa Clara, CA, USA). The images were obtained using microscopy (Leica Microsystems GmbH, Wetzlar, Germany), and the epithelial thicknesses were measured using Leica Application Suite X software (Leica, Microsystems GmbH).

### 4.11. Statistical Analysis

All results are presented as the mean ± standard error of the mean (SEM). The Mann-Whitney U test was used for comparisons between the two groups. Statistical analyses were conducted using PASW statistics Version 24 (IBM SPSS Statistics for Windows, Armonk, NY, USA), and *p* < 0.05 was considered significant.

## 5. Conclusions

We demonstrated that combination therapy with LTP, NPWT, and MSCs could effectively modulate inflammation by regulating TNF-α expression, induce angiogenesis by increasing VEGF, and promote tissue granulation by increasing α-SMA and type I collagen expression, which ultimately accelerated wound healing. These findings suggest the potential usefulness of these treatments in a clinical setting and indicate that further research may lead to significant breakthroughs.

## Figures and Tables

**Figure 1 ijms-21-03675-f001:**
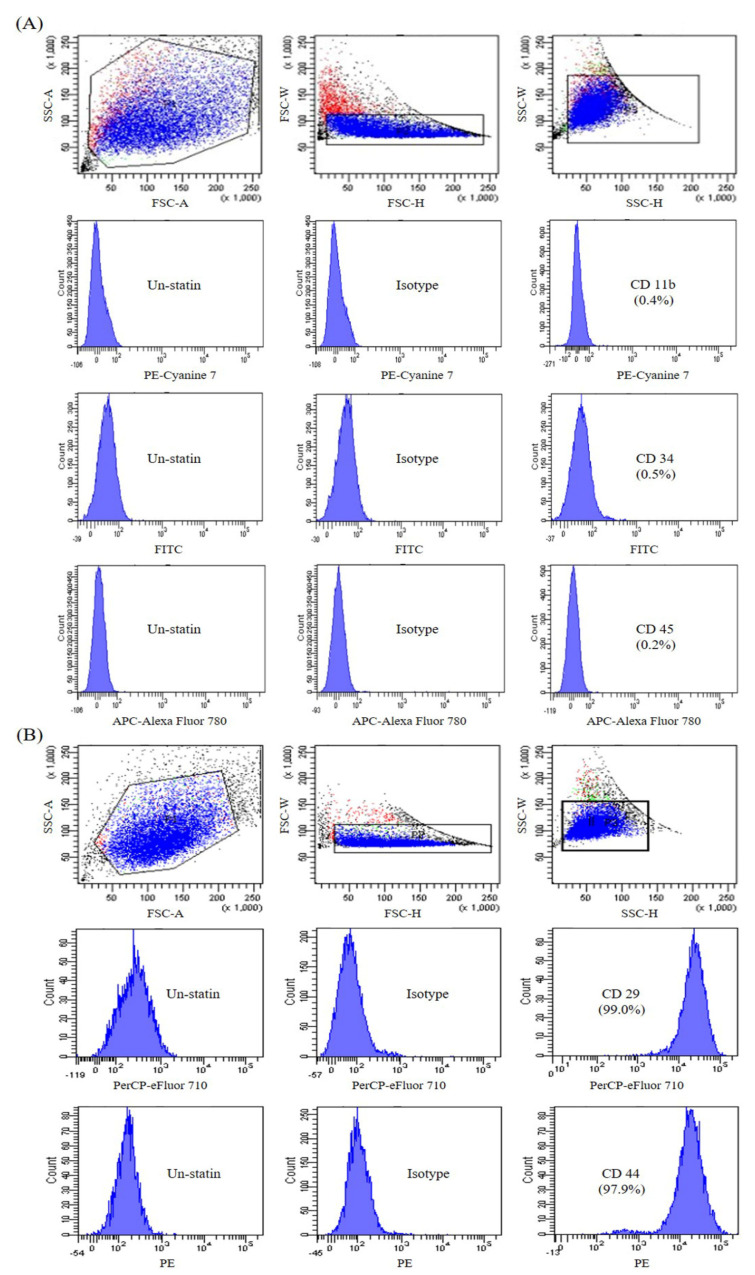
Fluorescence-activated cell sorting (FACS) analyses of mesenchymal stem cell MSC)-conjugated control isotype IgG or antibodies against indicated cell-surface proteins. MSCs were negative for CD11b, CD34, and CD45 (**A**) and positive for CD29 and CD44 (**B**).

**Figure 2 ijms-21-03675-f002:**
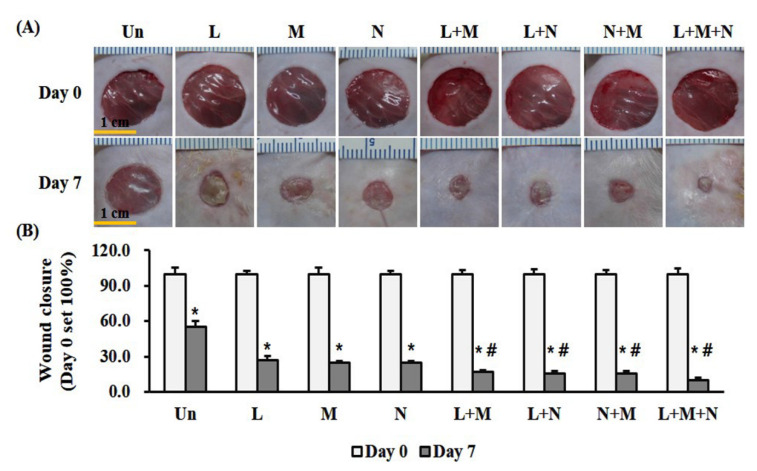
Mono- and combination therapy-accelerated wound healing in an ICR mouse full-thickness skin wound model. (**A**) Representative images of full-thickness wounds after mono- or combination therapy for 7 days. Scale bar = 1 cm. (**B**) Quantification of wound closure. Wound size was expressed as a percentage relative to the initial wound surface area (WSA), which was indicated as 100%. Data are mean ± SEM (*n* = 5). * *p* < 0.05 vs. the control, and ^#^
*p* < 0.05 vs. the monotherapy group at Day 7. LTP, low-temperature plasma; NPWT, negative pressure wound therapy; MSC, bone marrow mesenchymal stem cell; and Un, control.

**Figure 3 ijms-21-03675-f003:**
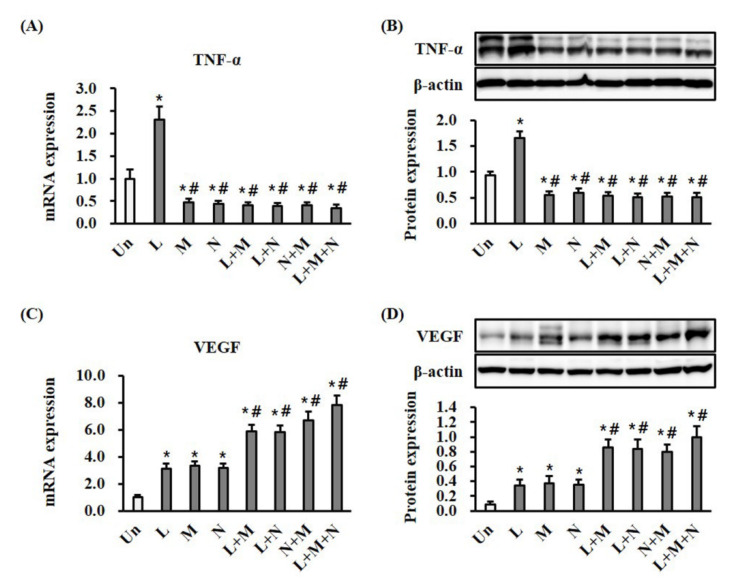
Mono- and combination therapy-modulated expression of TNF-α and VEGF in wound tissues. (**A**) Results of RT-qPCR analysis of TNF-α in wound tissue after treatment for 7 days. The mRNA expression levels are displayed as a fold change with respect to the levels in the control by the 2^−ΔΔCt^ method. * *p* < 0.05 vs. the control group. ^#^
*p* < 0.05 vs. the control and LTP monotherapy groups. (**B**) Results of Western blotting analysis of TNF-α in wound tissue. The protein expression levels were normalized with β-actin. * *p* < 0.05 vs. the control group, and ^#^
*p* < 0.05 vs. the comparable mono-treated groups. (**C**) Results of RT-qPCR analysis of VEGF in wound tissue. The mRNA expression levels are displayed as a fold change with respect to the levels in the control by the 2^−ΔΔCt^ method. * *p* < 0.05 vs. the control group, and ^#^
*p* < 0.05 vs. the control and comparable monotherapy groups. (**D**) Results of Western blotting analysis of VEGF in wound tissue. The protein expression levels are normalized with β-actin. * *p* < 0.05 vs. the control group, and ^#^
*p* < 0.05 vs. the control and comparable monotherapy groups. Data are mean ± SEM (*n* = 5). LTP, low-temperature plasma; NPWT, negative pressure wound therapy; MSC, bone marrow mesenchymal stem cell; and Un, control.

**Figure 4 ijms-21-03675-f004:**
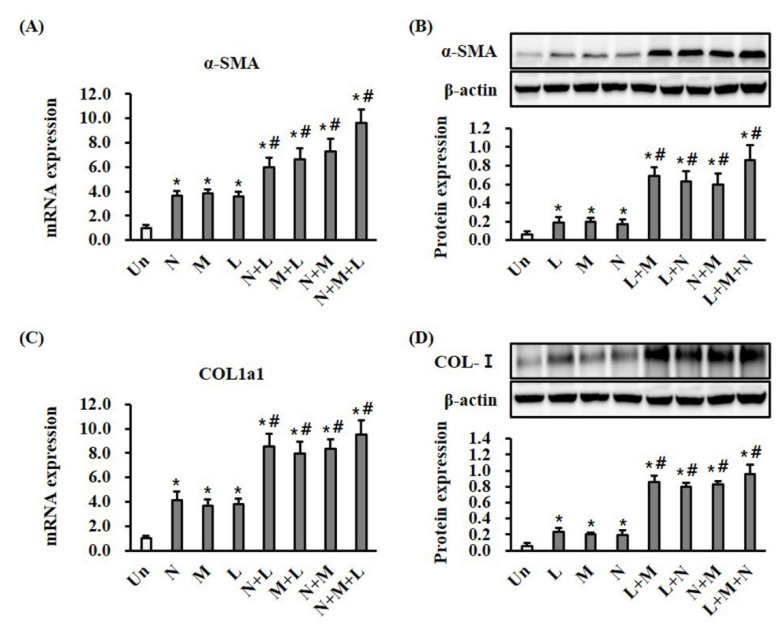
Mono- and combination therapy increased α-SMA expression and collagen deposition in wound tissues. (**A**) Results of RT-qPCR analysis of α-SMA in wound tissue after treatment for 7 days. The mRNA expression levels are displayed as a fold change with respect to the levels in the control by the 2^−ΔΔCt^ method. * *p* < 0.05 vs. the control group, and ^#^
*p* < 0.05 vs. the control and comparable monotherapy groups. (**B**) Results of Western blotting analysis of α-SMA in wound tissue. The protein expression levels were normalized with β-actin. * *p* < 0.05 vs. the control group, and ^#^
*p* < 0.05 vs. the control and comparable monotherapy groups. (**C**) Results of RT-qPCR analysis of type I collagen in wound tissue. The mRNA expression levels are displayed as a fold change with respect to the levels in the control by the 2^−ΔΔCt^ method. * *p* < 0.05 vs. the control group, and ^#^
*p* < 0.05 vs. the control and comparable monotherapy groups. (**D**) Results of the Western blotting analysis of type I collagen in wound tissue. The protein expression levels were normalized with β-actin. * *p* < 0.05 vs. the control group, and ^#^
*p* < 0.05 vs. the control and comparable monotherapy groups. Data are mean ± SEM (*n* = 5). LTP, low-temperature plasma; NPWT, negative pressure wound therapy; MSC, bone marrow mesenchymal stem cell; and Un, control.

**Figure 5 ijms-21-03675-f005:**
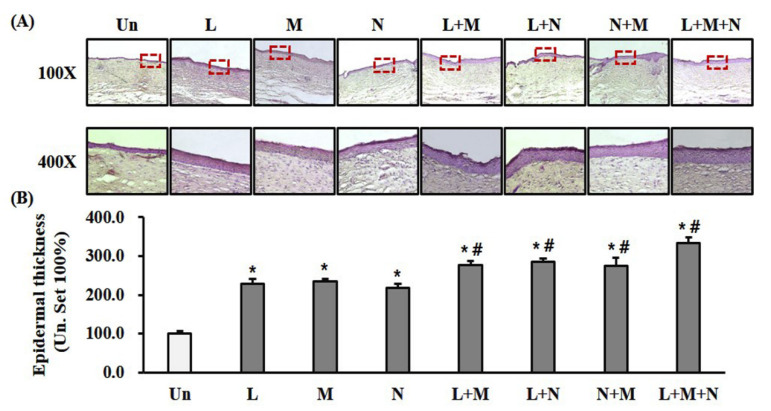
Mono- and combination therapy-enhanced re-epithelialization. (**A**) Representative images showing H&E staining of completely re-epithelialized wounds on day 21 after wounding. Scale bar = 100 μm at 100× and 25 μm at 400×. (**B**) Quantification of epidermal thickness. The epidermal thickness was expressed as a percentage relative to the control (control set at 100%). Data are mean ± SEM (*n* = 5); * *p* < 0.05 vs. the control group, and ^#^
*p* < 0.05 vs. the control and comparable monotherapy groups. LTP, low-temperature plasma; NPWT, negative pressure wound therapy; MSC, bone marrow mesenchymal stem cell; and Un, control.

**Table 1 ijms-21-03675-t001:** Flow cytometry antibodies.

Antibody	Cat. no.	Brand
CD11b-PE-Cyanine 7	11-0341	eBioscience
PE-Cyanine 7 (Isotype)	25-4031	eBioscience
CD29-PerCP-eFluor710	46-0291	eBioscience
PerCP-eFluor710 (Isotype)	46-4888	eBioscience
CD34-FITC	11-0341	eBioscience
FITC (Isotype)	11-4321	eBioscience
CD44-PE	12-0441	eBioscience
PE (Isotype)	12-4031	eBioscience
CD45-APC-eFluor 780	47-0451	eBioscience
APC-eFluor 780 (Isotype)	47-4031	eBioscience

**Table 2 ijms-21-03675-t002:** Real-time PCR primer sequences.

Gene	Forward (5′ → 3′)	Reverse (5′ → 3′)
*TNF*	ACTCCAGGCGGTGCCTATGT	GTGAGGGTCTGGGCCATAGAA
*VEGFA*	ACAGAGTATTTGCGCTCCGAA	TGGTTGGAACCGGCATCTTTA
*ACTA2*	CCGACCGAATGCAGAAGGA	ACAGAGTATTTGCGCTCCGAA
*COL1A1*	ATGTTCAGCTTTGTGGACCTC	CTGTACGCAGGTGATTGGTG
*GAPDH*	CATGGCCTTCCGTGTTCCTA	TGTCATCATACTTGGCAGGTTTCT

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
