# Peer review of "Effect of Combining Low Temperature Plasma, Negative Pressure Wound Therapy, and Bone Marrow Mesenchymal Stem Cells on an Acute Skin Wound Healing Mouse Model"

_ijms, 2020, doi:10.3390/ijms21103675_

Round 1
Reviewer 1 Report
The combination of plasma, negative pressure, and MSC for improved wound healing is very original. The conclusions are supported by the results, and I have only a few minor comments before recommending the manuscript for publication:
- fig 1 should be improved. first, its obvious that the first six figures and the last four figures were layouted together, and the labeling is very poor. why don't you write "unstained", "isotype", and "stained" over each "column" of staining? graphs should be aligned and have the same size. and, if you chose to show flow cytometry data, show 1-2 dot plots above the histograms to exemplify your fsc/ssc gating - its obvious that a dead cell stain was missing, so there must have been a logical input gate that went into the histograms.
- a study limitation should be amended to the discussion. the wound model at the back of the animals is largely related to muscle-contraction-mediated healing, and not re-epithelialization as in humans. this needs to be noted.
- ethical approval number is missing but needs to be included!
- comment on your negative pressure therapy: did you really have the mice every day for 4h straight in anesthesia for the negative pressure therapy? seems a bit long. or did you fix the animals otherwise without drugs? you had to fix or sedate them for the treatment, i suppose?
Author Response
Dear Reviewers,
We would like to thank the reviewers for the careful review of our manuscript and their helpful suggestions. The suggestions have helped us improve the quality of our manuscript. Additionally, we would like to thank them for their kind words concerning our manuscript’s topic, organization, and suitability for publication.
Below, we have responded in a point-by-point manner to the comments made by the reviewers.
Reviewer 1
The combination of plasma, negative pressure, and MSC for improved wound healing is very original. The conclusions are supported by the results, and I have only a few minor comments before recommending the manuscript for publication:
- Fig 1 should be improved. First, its obvious that the first six figures and the last four figures were layouted together, and the labeling is very poor. why don't you write "unstained", "isotype", and "stained" over each "column" of staining? graphs should be aligned and have the same size. and, if you chose to show flow cytometry data, show 1-2 dot plots above the histograms to exemplify your fsc/ssc gating - its obvious that a dead cell stain was missing, so there must have been a logical input gate that went into the histograms.
We have revised our manuscript according to the reviewer's suggestion and believe that replacing the original Figure 1 with a new Figure 1, which illustrates our FACS results, better conveys our intended meaning.
- A study limitation should be amended to the discussion. The wound model at the back of the animals is largely related to muscle-contraction-mediated healing, and not re-epithelialization as in humans. This needs to be noted.
Thank you for your instruction. We have discussed this limitation by referring to structural and functional differences in wound healing between humans and lower mammals (lines 224-229).
- Ethical approval number is missing but needs to be included!
We have revised our ‘Material and Method’ section to include this number (line 312), and thank the reviewer for their keen observation.
- Comment on your negative pressure therapy: did you really have the mice every day for 4h straight in anesthesia for the negative pressure therapy? seems a bit long. or did you fix the animals otherwise without drugs? you had to fix or sedate them for the treatment, i suppose?
Although this was a fairly long duration of NPWT, standard clinical practices indicate that when NPWT is used for wound healing, it should be performed every day for ~2–4 h (lines 180-183). We chose the maximum duration in the hope of obtaining optimal therapeutic effects. To accomplish this, we put the mice under anesthesia for the 4 h treatment period each day, following pre-treatment with Rompun (0.04 mL/kg) for sedation.
Reviewer 2 Report
The work by Cui et al. try to address the effects that combining different well established methods for improving wound healing may provide. They obtain clear results on that purpose, however they fail in providing meaningful description about the underlying mechanisms providing such results, mainly relying in very short explanations backed by few references. As such, although the paper is generally well structured and clear in its results, I find it too shallow and a bit out of scope for being published in IJMS. For this I recommend major revision or looking for another journal better suited for this piece of research.
Some comments on precise aspects to improve:
- Title fails to address the fact that the model used is an acute wound healing one.
- Same in abstract.
- Line 41, word "perplexing" may result difficult to put in context.
- At the end of the introduction, a precise description of the model used looks necessary.
- Section 2.1 is too brief and lacks description of the markers used, neither is found in captions of figure 1. The relevance for the study of such markers is just marginally addressed in the discussion section.
- In discussion section:
- Too short and brief, sometimes jumping from one issue to another without clear link. Reading it one can get the feeling that more care is needed.
- Working around the molecular details involved, as little experimental evidence is shown regarding some mechanisms proposed, at least sound support from literature is to be expected while discussing the limitations of such support. This is somewhat lacking.
- Lines 186-188, better writing is needed. The participation of Nrf2 should be put in the context of acute wound healing. Also consider that TGF-b has been to show many roles in wound healing far more complex than what is proposed.
- Data not shown about TSG6 is in dispute with the journal policy. This data should be included and properly addressed.
- Line 198 is missing appropriate references if available
- Lines 198-210, paragraph is very confusing.
- Line 211-215, were wound contraction is addressed... but no mention through the text about the limitations that using animal models carry for studying wound healing. Animals display well developed panniculus carnosus contributing to wound contraction. This tissue layer is missing in humans and crucial divergences emanate from this fact, as widely showed in literature.
- At the end of the discussion section there is mention about the potential of the application of the results of this work for chronic wound healing, however nor the models neither the evidence shown supports such conclusion. This should be either removed or clarified.
- Methods:
- Mice strain should be properly identified in each section were used.
- Setion 4.8 states the use of ß-actin as a reference housekeeping gene. Since several research pieces have shown the involvement of actin in the regulation of cell processes involved wound healing I advice explore using a different reference gene such as GAPDH.
Author Response
Dear Reviewers,
We would like to thank the reviewers for the careful review of our manuscript and their helpful suggestions. The suggestions have helped us improve the quality of our manuscript. Additionally, we would like to thank them for their kind words concerning our manuscript’s topic, organization, and suitability for publication.
Below, we have responded in a point-by-point manner to the comments made by the reviewers.
Reviewer 2
The work by Cui et al. try to address the effects that combining different well established methods for improving wound healing may provide. They obtain clear results on that purpose, however they fail in providing meaningful description about the underlying mechanisms providing such results, mainly relying in very short explanations backed by few references. As such, although the paper is generally well structured and clear in its results, I find it too shallow and a bit out of scope for being published in IJMS. For this I recommend major revision or looking for another journal better suited for this piece of research.
Some comments on precise aspects to improve:
- Title fails to address the fact that the model used is an acute wound healing one. Same in abstract.
Thank you for your suggestion. We have updated our title to reference acute wound healing.
- Line 41, word "perplexing" may result difficult to put in context.
We agree with the reviewer’s observation and have rewritten this section to more directly capture our meaning (lines 36-37).
- At the end of the introduction, a precise description of the model used looks necessary.
We have revised line 47 to explicitly state the model to which we referred.
- Section 2.1 is too brief and lacks description of the markers used, neither is found in captions of figure 1. The relevance for the study of such markers is just marginally addressed in the discussion section.
We have expanded section 2.1 to include more information regarding our use of markers as a method for determining MSC purity (lines 70-76). Additionally, we have substantially increased our explanation of the relevance of these markers throughout the discussion section.
- In discussion section: Too short and brief, sometimes jumping from one issue to another without clear link. Reading it one can get the feeling that more care is needed.
Thank you for your helpful feedback. We have substantially increased the breadth and depth of our discussion section, which has resulted in increased length and clarity.
- Working around the molecular details involved, as little experimental evidence is shown regarding some mechanisms proposed, at least sound support from literature is to be expected while discussing the limitations of such support. This is somewhat lacking.
Further experimentation focused on molecular mechanisms was beyond the scope of our research. However, a large portion of our expanded discussion section focused on the mechanisms underlying our findings through references to complementary research. We believe that this has clarified the connections between mechanisms and clinical applications.
- Lines 186-188, better writing is needed. The participation of Nrf2 should be put in the context of acute wound healing. Also consider that TGF-b has been to show many roles in wound healing far more complex than what is proposed.
After careful consideration, we decided to remove the information in the discussion concerning Nrf2. We agree with the reviewer that this link was not justified by our findings. We agree that the role of TGF-β1 in wound healing is complex. Given this, we have considerably enhanced the information presented concerning TGF-β1, in our discussion section, with special attention to its relevance in lines 207-223.
- Data not shown about TSG6 is in dispute with the journal policy. This data should be included and properly addressed.
We have included the data concerning TNF-α and TSG-6 after LTP treatment as supplementary results.
- Line 198 is missing appropriate references if available
We have included the appropriate references.
- Lines 198-210, paragraph is very confusing.
We have revised this paragraph to better convey our intended meaning.
- Line 211-215, were wound contraction is addressed... but no mention through the text about the limitations that using animal models carry for studying wound healing. Animals display well developed panniculus carnosus contributing to wound contraction. This tissue layer is missing in humans and crucial divergences emanate from this fact, as widely showed in literature.
Thank you very much for your observation. We have addressed this limitation in lines 226-228.
- At the end of the discussion section there is mention about the potential of the application of the results of this work for chronic wound healing, however nor the models neither the evidence shown supports such conclusion. This should be either removed or clarified.
We have rephrased this section to better convey our intended meaning (lines 215-217).
- Methods: Mice strain should be properly identified in each section were used.
We have now clarified throughout the manuscript that the ICR strain of mice was used for all our experiments.
- Setion 4.8 states the use of ß-actin as a reference housekeeping gene. Since several research pieces have shown the involvement of actin in the regulation of cell processes involved wound healing I advice explore using a different reference gene such as GAPDH.
We have addressed this according to the reviewer's suggestion and revised our RT-qPCR results using GAPDH as a reference.
Round 2
Reviewer 2 Report
The work by Cui et al. try to address the effects that combining different well established methods for improving wound healing may provide. They obtain clear results on that purpose, however they fail in providing meaningful description about the underlying mechanisms mainly relying in literature support. Nevertheless, the authors had made an effort to fix crucial aspects that needed to be clarified.
As such, the paper is generally well structured and clear enough in its results to be worth of publication; yet I must observe that some more ambition at the molecular level would benefit the work.
Also, some grammar corrections and clarifications in the text are still need. For this I recommend accept after minor revision (corrections to minor methodological errors and text editing).
Some comments on precise aspects to improve (on revised text with all changes):
- Title might be missing "an" between "on" and "acute"
- Lines 24-25-26 see wrong formating
- Line 194 should be "promoted" instead of "promotes"
- Phrase from lines 202-204 is confusing, please amend
- Consider phrase in line 267-269, revise recent works questioning TGF-ß levels in chronic wounds https://doi.org/10.3390/cells9020306
- Paragraph in lines 276-281, in my opinion, it would better fit within the closing paragraph. Optimal position might lay in line 302 before the closing sentence.
- Line 278, "our results may not be valid" is rather too restrictive, use more precise phrasing like "limitations apply for translating our results".
- Line 303, "is" implies the outcomes are already validated in humans; this should change since the combination therapy and its outcome in humans is still a potential prospect.
- IMPORTANT ISSUE: supplementary material is missing from the revised manuscript.
Author Response
Dear reviewers,
Once again, we would like to thank the reviewers for their work and comments on our manuscript. We are thankful for their kind words considering our article interesting, well-organized and suitable for publication.
In the following lines we will reply to each of the comments made by the reviewers.
Reviewer 2
Title might be missing "an" between "on" and "acute"
We revised it according to reviewer's suggestion.
Lines 24-25-26 see wrong formatting
We revised it according to reviewer's suggestion, at line 24-25.
Line 194 should be "promoted" instead of "promotes"
We revised it according to reviewer's suggestion, at line 169.
Phrase from lines 202-204 is confusing, please amend
We revised it according to reviewer's suggestion, at line 203-204.
Consider phrase in line 267-269, revise recent works questioning TGF-ß levels in chronic wounds https://doi.org/10.3390/cells9020306
We revised it according to reviewer's suggestion, at line 211-213.
Paragraph in lines 276-281, in my opinion, it would better fit within the closing paragraph. Optimal position might lay in line 302 before the closing sentence.
We revised it according to reviewer's suggestion, at line 237-243.
Line 278, "our results may not be valid" is rather too restrictive, use more precise phrasing like "limitations apply for translating our results".
We revised it according to reviewer's suggestion, at line 239-240.
Line 303, "is" implies the outcomes are already validated in humans; this should change since the combination therapy and its outcome in humans is still a potential prospect.
We revised it according to reviewer's suggestion, at line 244-245.
IMPORTANT ISSUE: supplementary material is missing from the revised manuscript.
We added it to the "supplementary results" file as recommended by the reviewer.